

# Gridded Satellite (GridSat) GOES and CONUS data

Kenneth R. Knapp[1] and Scott L. Wilkins[2]

[1]NOAA/NESDIS/National Centers for Environmental Information, Asheville, NC 28801, USA
[2]Cooperative Institute for Climate and Satellites – North Carolina (CICS-NC), North Carolina State University, Asheville, NC 28801, USA

*Correspondence to*: Kenneth R. Knapp (Ken.Knapp@noaa.gov)

**Abstract.** The Geostationary Operational Environmental Satellite (GOES) series is operated by the U. S. National Oceanographic and Atmospheric Administration (NOAA). While in operation since the 1970s, the current series (GOES 8-15) has been operational since 1994. This document describes the Gridded Satellite (GridSat) data, which provides GOES data in a modern format. Four steps describe the conversion of original GOES data to GridSat data: 1) temporal resampling to produce files with evenly spaced time steps, 2) spatial remapping to produce evenly spaced gridded data (0.04° latitude), 3) calibrating the original data and storing brightness temperatures for infrared channels and reflectance for the visible channel, and 4) calculating spatial variability to provide extra information that can help identify clouds. The GridSat data are provided on two separate domains: GridSat-GOES provides hourly data for the Western Hemisphere (spanning the entire GOES domain) and GridSat-CONUS covers the contiguous U.S. (CONUS) every 15 minutes.

Dataset reference: doi:10.7289/V5HM56GM

## 1 Introduction

NOAA has used the Geostationary Operational Environmental Satellite (GOES) series since 1975 to monitor weather. While the series name has remained the same, the satellites providing the data have seen step-wise increases in capabilities. The original system was a spin-stabilized system. The system received a significant upgrade in 1994 with the launch of GOES-8 (the GOES I-M series), providing a 5-channel imager (Menzel and Prins, 1996). The most recent launch (in 2016) of GOES-16 is another substantial increase in capabilities (Schmit et al., 2016). The satellites have been maintained at two primary locations: GOES-West at 135° West which observes the Eastern Pacific and western North America and GOES-East at 75° West, providing coverage of North and South America and much of the Atlantic Ocean.

Gridded Satellite (GridSat) is produced as a means to facilitate access to GOES satellite data. There are two domains provided: a 15 minute domain provided over the contiguous U.S. called GridSat-CONUS and an hourly domain that spans the Western Hemisphere to cover the entire GOES domain, called GridSat-GOES. In the following, we use the term GridSat-GOES to describe both these datasets, since their only difference is the temporal resolution and spatial coverage.



## 1.1 Purpose for Gridded GOES data

Since the launch of the GOES I-M series in 1994, worldwide computing capabilities have significantly improved and changed the landscape of how the data are transmitted, accessed, and used. When first launched, most GOES processing was conducted using the Man computer Interactive Data Access System (McIDAS) and data were transferred via satellite broadcast or high

density tapes. Thus, data were used primarily for weather research and forecasts. About 25 years later, the data can be accessed with a variety of computers and software, primarily transferred over the Internet and used for a myriad of purposes. Thus, the demand has significantly increased while the support of GOES formats has remained stagnant. Prior to GridSat-GOES, the data were available from the archive as:

- Raw GVAR data – A very complex binary format intended for transmission of satellite data via satellite uplink and

downlinks. This can be read by a very limited set of software. The target audience are satellite experts with access to large computing resources.

- AREA format – A format developed for MCIDAS to process satellite data. This is well documented, but still requires a thorough knowledge of satellite data to extend to other applications. The target audience are those with access to MCIDAS software (e.g., academia).

- netCDF format – The netCDF file provided by the archive does not follow accepted standards in storing data and it provides no information on converting values to reflectance/brightness temperatures. There is also no obvious way to renavigate imagery. Therefore, while the target audience may be scientific users, these issues make the data provided by the archive less useful for that audience.

- TIFF/JPEG – These formats are solely for imaging. The data provided have map overlays. The target audience are

the general public looking for satellite images for qualitative analysis, not quantitative studies.

What is lacking is a format that can be used by the general scientific community, such that it can easily be incorporated into other processes and used for analysis in a wide range of applications.

Table 1 contrasts the access to GOES data via MCIDAS Area files and the GridSat-GOES formats. The AREA file format is fully supported by one software application, the navigation (outside of McIDAS) requires more than 1000 lines of

code and the calibration information (outside of McIDAS) is not directly available in the AREA files, but requires accessing satellite specific coefficients from another source. Lastly, the temporal resolution of the files can vary from 1 minute to 30 minutes, with a complex scan schedule that varies depending on the scan mode selected for the day (e.g., when severe weather is expected, forecasters can request 1 minute rapid scans of a region, at the expense of other regional scans). Conversely, the GridSat-GOES format provides calibrated and mapped data that has been temporally sampled to the nearest 15 minutes (for

CONUS) and 1 hour (for GOES). The format – netCDF4 – is supported by Unidata, which provides information on how to use netCDF in multiple languages and applications.



## 1.2 Relationship to other available datasets

GridSat-GOES and –CONUS are most similar to two other datasets: the Global Merged IR dataset (Janowiak et al., 2001) and the GridSat-B1 Climate Data Record (CDR) (Knapp et al., 2011). Various details of the datasets are compared in Table 2. The global merged data has provided information for precipitation processing since the beginning of global meteorological geostationary satellite coverage in 1998, but only provides information from one channel: the infrared window. The GridSat CDR expands on this by providing intercalibrated IR brightness temperatures with a longer period of record (1980-present). The GridSat-GOES datasets focus on providing access to complete GOES data in a new, updated format. It provides access to all GOES channels. Since it does not expand to other satellites, it is mostly limited to the Western Hemisphere.

## 2 GOES Data processing

The effort to produce GridSat data read and processed GOES Area files retrieved from NOAA CLASS (Comprehensive Large Array-data Stewardship System). In so doing, we ordered and downloaded 294 TBs of data in about 10 million files. This represents the entire archive of GOES 8-15 data for 1994-2016. Processing these AREA files required several steps in order to produce the final GridSat products. The first step was to resolve the issue of duplicate files in the archive. This occurs when multiple files are available for one scan time. Next, simplified the scan times were selected. While the GOES scan schedule is a complex system of various scan schedules taking image scans (i.e., regions) at different times, the GridSat system provides grids at regular times. Image scans are then navigated, which involves determining the location of each pixel and, conversely, determining which image pixel is closest to a given Earth location. Once navigate, the scans are converted from digital numbers with a 10-bit depth (hence they range from 0-1023) to geophysical units, either reflectance or brightness temperature, a process called calibration. Lastly, two other variables which provide information on the spatial uniformity of the images are calculated and stored in the resulting GridSat file. More details on these procedures are provided below.

## 2.1 Resolving duplicate files

Files were originally archived and stored on tapes. Each time a tape is read, there is the possibility that the file has slightly different contents. When this occurs, both files are retained and are differentiated by a trailing hyphen and number (e.g., "-1").

Duplicate files are prevalent in the GOES archive. They exist for 1994 through 2003 and can be up to 30% of the files for any given year. Because some files have more than one duplicate; it is not unusual to find files that are repeated 3 or 4 times. The most repeated file is "goes08.1995.060.174513" [from GOES-8 on Mar. 1, 1995 at 17:45:13 UTC] which has 6 files for the single scan. The total number of extra files in the archive exceeds 64,000 files; for context, that is more than the number of GOES files archived for 1997. In essence, there is an extra satellite year in the archive.

The reprocessing effort set up an objective algorithm to reconcile these duplicate files. Files were compared scan line by scan line. First, files that were identical were resolved simply by renaming or removing files. Where scan lines differed, the scan lines that were more highly correlated with neighbouring scan lines were selected. This led to selecting one of the multiple



files as the best, however some cases resulted in a completely new file (when best scans were selected from different files). This process resolved all duplicated files.

## 2.2 Temporal Sampling

There are generally about 120 GOES scans per day; the exact number varies by scan schedule. Given the wide variety of scans and times, the GridSat-GOES and CONUS sectors operate at optimal 60 minute and 15 minute intervals, respectively. That is, a sector that begins at 17:45 will be included in the 17:45 CONUS grid as well as the 18:00 GOES grid. Thus, users do not need to be aware of the scan schedule, or understand where each is taken, in order to access and process GridSat-GOES data.

Furthermore, the hourly processing allows GridSat-GOES to merge images from various scans. The GOES scan schedule provides a full disk scan once every three hours. However, the various scans in the intermediate times, when merged, produce something that can approximate full disk coverage. Figure 1 provides a demonstration of this capability. Fig. 1a shows an 18:00Z image, for which a full disk scan was available. However, in Fig. 1b, the sectors from multiple separate scans (Northern Hemisphere-Extended, Southern Hemisphere, etc.) are merged on the same grid, thus providing a nearly full disk image.

The GridSat-GOES dataset provides a variable (called delta_time) that allows the calculation of the actual scan time of each pixel by storing the difference in time between the optimal image time (17:45 and 18:00 in the previous example) and the actual scan time of the image. This allows for calculation of accurate solar illumination angles.

## 2.3 Navigation

Once a grid time is selected, the image pixels nearest each grid cell are selected through sampling. No averaging of the observations are performed. Sampling maintains the pixel bit depth, radiative characteristics and spatial resolution. Data were navigated following the algorithms provided for GOES data by (National Environmental Satellite Data and Information Service (NESDIS), 1998). Some navigation errors will be present in the remapped data from the original data;  navigation accuracy for GOES is about 4 km during daylight and 6 km at night (Ellrod et al., 1998).

## 2.4 Calibration

Infrared channel data are calibrated following documentation from the NOAA STAR web page (Weinreb et al., 2017; Weinreb et al., 1997). Scan lines are calibrated separately and converted to a brightness temperature. The brightness temperatures are stored in 2-byte integers using data packing (e.g., CF convention scale_factor and add_offsets). Brightness temperature noise is generally less than 0.25 K (Ellrod et al., 1998) which was later evaluated by Wang et al. (2011) who found errors at night on the order of 0.15 K.

There is no on-board visible calibration reference for any GOES satellite prior to GOES 16, so some correction is necessary given the decay of the instrument gain (Knapp and Vonder Haar, 2000). Visible channel data are calibrated in GridSat-GOES following Inamdar and Knapp (2015).



## 2.5 Visible and Infrared Window variability

Lastly, two extra variables are provided: IR and visible channel variability. For each pixel that was mapped to a grid cell, the spatial variability is provided. The spatial variability is calculated as the standard deviation of the 3x3 pixels centered on the selected pixel. This is calculated at the original satellite resolution. Thus, for the visible channel, which is available at 1 km pixels, it provides a measure of sub-grid cell variability. For the IR, whose original resolution is near 4km, the calculation provides a measure of the grid cell variability. Both of these measures are related to cloud cover and can be used to determine the presence of clouds (Rossow and Garder 1993).

## 3 Future Plans

Many possible improvements exist. Future improvements will depend on user needs. Possible improvements include:

- Expansion to 1980-1994- GridSat-GOES is currently limited to the GOES I-M series of satellites (GOES 8 through GOES-15). The dataset can be expanded back in time to provide information from the previous series (GOES 1-7). While it would have fewer channels, it could maintain the same temporal and spatial resolutions.

- Expansion to GOES-16 – The newest series of GOES increases the number of channels to 16, increases the spatial resolution to 2 km for most channels and the temporal resolution of full disk scans to half hourly. Nonetheless, there will continue to be demand for this long-term series at a reduced volume. Expanding GridSat-GOES to GOES-16 could provide a solution for users needing large spans of time.

- Cloud Information – Including cloud information (probability, temperature, optical thickness, etc.) is possible and can be provided in future versions. While increasing the volume of the data, it would increase the usefulness to many users.

- Increase spatial resolution – Visible data is available at 1 km spatial resolution and there has been some demand for that. However, it would significantly increase the volume (thus decreasing the usefulness of the data). One viable option would be to include the 1 km channel data only for the CONUS sector.

- Merge GOES-East/West – Data are presently provided separately for GOES-East and -West. It is possible to merge the projections to one value, but doing so would impact data usefulness. In particular, the visible channel would be affected by including satellites in the same scene that have different viewing geometries.

- Climate quality calibration – Presently, we are applying the operational brightness temperature corrections. There are higher quality calibrations that could be derived. While the visible channel calibration already includes values that provide longer term stability, improvements could be made.

This is a short list of changes that could be made to further improve the utility of the data. User feedback will be used to prioritize these tasks and resources applied to those that are in highest demand.





## 4 Data Availability

The data are available from the following links:

*GridSat-CONUS*         *https://www.ncei.noaa.gov/data/gridsat-goes/access/conus*

*GridSat-GOES*           *https://www.ncei.noaa.gov/data/gridsat-goes/access/goes*

The web page https://www.ncdc.noaa.gov/gridsat/ provides additional GridSat-GOES information and allows user registration. Table 3 provides a list of files (for GridSat-GOES) provided by each satellite for each year. This shows the progression of the GOES-East position from GOES-8 to -12 to -13; a similar progression is apparent for GOES-West. There are also numerous GridSat-GOES files for GOES 14, however, it was not listed in Table 3 since it never assumes a role as either GOES-East or -West. The entire GridSat-GOES period of record for all satellites is 24TB, which is a factor of 10 smaller than the full

resolution archive (342 TB). The GridSat-CONUS period of record totals about 5 TB of data.

## 4.1 Variables

The following variables are provided in GridSat-GOES and -CONUS netCDF files. They are listed with a summary of how they could be used. A sample set of images is provided in Figure 2 for GridSat-CONUS.

- Lat – The latitude coordinate variable (in ° North). This provides the ability to map the data.
- Lon – The longitude coordinate variable (in ° West), also provides the ability to map the data.
- Time – Optimal time for the GridSat data (in units of 'days since 1/1/1970 00:00:00 UTC').
- Lat_bounds – From the CF convention, these provide the spatial bounds of the grid cells used.
- Lon_bounds – From the CF convention, these provide the spatial bounds of the grid cells used.
- Time_bounds – From the CF convention, these provide the temporal bounds of the temporal sampling.
- Satlat – The latitude of the sub-satellite point (in ° North). This can be used in conjunction with satlon to calculate the viewing angles (e.g., satellite view zenith and azimuth angles).
- Satlon – The longitude of the sub-satellite point (in ° West).
- Satrad – Distance (in km) of the satellite from the center of the Earth.
- Filename – Filenames for the source archive AREA files included in the GridSat file, to provide lineage.
- Delta_time – Provides the difference of the actual scan time (in minutes) from the optimal grid time, used for calculations of solar geometry.
- Ch1 – Reflectance (or scaled radiance) for the 0.6 um band. The visible channel provides observations of clouds, Earth's surface, and aerosols.
- Ch2 – Brightness temperature of the 3.75 um channel. This wavelength measures both reflected sunlight as well as
the Earth's radiance and provides information on cloud microphysics and other surface information.



- Ch3 – Brightness temperature of the 6.7 um channel, also called the water vapor channel. The channel provides information on the distribution and movement of water vapor in the upper atmosphere. The atmosphere at this wavelength is opaque and the channel rarely sees the surface.

- Ch4 – Brightness temperature of the 11 um channel, generally called the IR window channel. It provides information on the radiant temperature on the surface or cloud top.

- Ch5 – Brightness temperature of the 12 um channel. This channel provides information on the amount of atmospheric contamination in the 11um, which is why it is often called the split window. This channel is empty for GOES 12-15 satellites because it is not included on those satellites.

- Ch6 – Brightness temperature of the 13 um channel. This channel provides information on cloud cover and cloud height. This channel is empty for GOES-8-11 satellites.

- Ch1v – Spatial standard deviation of the channel 1 reflectance. This provides a measure of sub grid cell variability because the calculation is made on the original satellite data, then sampled to the grid cells. Useful in identifying clouds (Rossow and Garder 1993).

- Ch4v – Spatial standard deviation of the channel 4 (infrared window) brightness temperatures, useful for identifying clouds.

As mentioned in Sec. 3, it could be possible to expand this variable list as deemed necessary to meet user needs.

## 4.2 Coverages

The coverages of GridSat-GOES and –CONUS are shown in Figure 3. GridSat-GOES spans the Western Hemisphere; in fact, it extends into the Eastern Hemisphere, spanning 150° East to 5° East and spanning 75° South to 75° North. The GridSat-CONUS spans 125° to 65° West and 25° to 50° North. Both sectors are provided at 0.04°, which is approximately 4 km near the Equator.

## 4.3 Uncertainty

Menzel and Purdom (1994) provide information on GOES calibration accuracy based on pre-launch design. Post-launch analysis has deemed IR calibration accurate to within 0.15 to 0.25 K (Ellrod et al., 1998; Wang et al., 2011). The visible calibration was provided by Inamdar and Knapp (2015), where the uncertainty is about 3% (in scaled reflectance units).

## 4.4 Caveats

While the GridSat-GOES dataset is suitable for scientific and research use, some caveats should be considered prior to usage.

The IR calibration is not yet climate quality. We presently use the operational calibration without adjustment. There are, however, some issues that reduce the ability to use GridSat-GOES for climate use. For instance, the midnight black body calibration correction was implemented by NOAA in 2003. Therefore, IR calibration is lower quality near the satellite local



midnight prior to April 1, 2003. Also, the GSICS (Global Space-based Inter-Calibration System) community has worked on IR calibration adjustments (Goldberg et al., 2011) which are also not included in this initial version.

Users should also be aware of the limitations of the temporal and spatial sampling. The full resolution visible data are sampled to the 0.04° grid spacing. Similarly, full resolution temporal scans (e.g., 1-minute rapid scans) are used only when

they provide an observation closest to the optimal grid time. Therefore, users requiring high spatial or temporal sampling will need to access GOES data using another method.

Another limitation of the temporal sampling is the gaps that occur when data are not available. Three primary causes for the gaps are: scan schedules, eclipse keep-out zones, and archive gaps. The scan schedule causes gaps mostly to the CONUS grid when the satellite is scanning the full disk (which takes longer than 15 minutes). For example for GOES-East, there is no

18:00 GridSat-CONUS file since it is still doing a full disk scan that started at 17:45. These gaps generally occur at regular intervals and are usually limited to one missed CONUS grid. The keep-out zone refers to when the satellite is near the local midnight and has the potential to look directly at the Sun. Since this would permanently damage the satellite sensor and optics, a keep-out zone was implemented to reduce the possibility of damage. These are times near local midnight near the March and September equinoxes. The resulting gaps are daily and last a few hours. The GOES N-P series includes improved design that

allows for shorter keep-out zones. Another limitation is the archive itself. There are some gaps in the archive where scans exist in other archives for one reason or another. These gaps can be longer than those caused by the scan schedule, often exceeding a day.

A complete list of known caveats and issues is maintained on the GridSat-GOES website.

## 5 Conclusions

The GridSat-GOES and –CONUS datasets provide GOES data to a wide audience in a standard format. Since the tasks of calibrating and navigating have been performed during production, data access is greatly simplified for the user. The temporal granularity structure is also much simpler, using fixed grid times rather than the complex and variable heritage scan times. The data are then made available in the widely supported netCDF-4 format. Future improvements and versions will be driven by user requirements.

**Acknowledgments**

We would like to acknowledge Jessica Matthews and Anand Inamdar of CICS-NC for making this GOES reprocessing possible. Scott W. ordered and managed the GOES data downloaded from NOAA CLASS. Jessica M. and Anand. I. provided feedback on the manuscript, helped guide the dataset production, and acted as test data users. Lastly, this dataset would not have been produced without the computational support of CICS-NC. This work was supported by NOAA through the

Cooperative Institute for Climate and Satellites - North Carolina under Cooperative Agreement NA14NES432003.



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


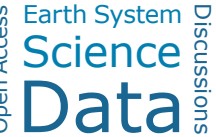

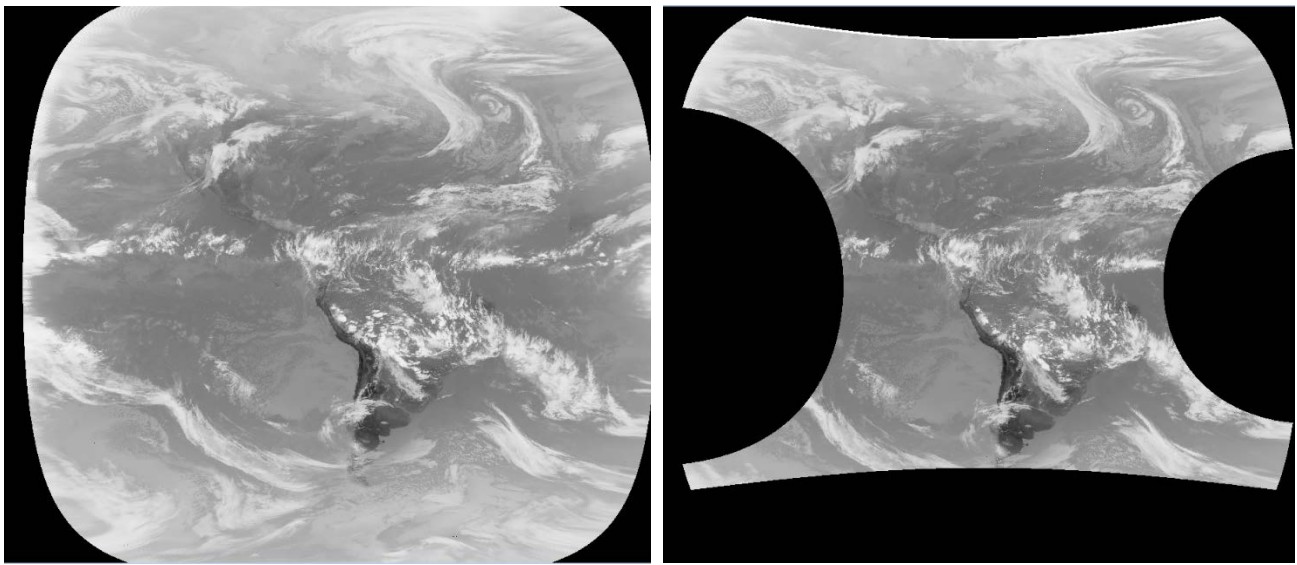

**Figure 1 – Sample of the GridSat-GOES images for 18:00 UTC and 19:00 UTC. A) 18:00 image provided by a full disk scan. B) 19:00 image is a combination of 3 separate scans.**





**Figure 2 – GridSat-CONUS sample for 19 Aug 1999 at 17:45 UTC showing all 8 channels. Since this is GOES 8, there is no Channel 6 data.**




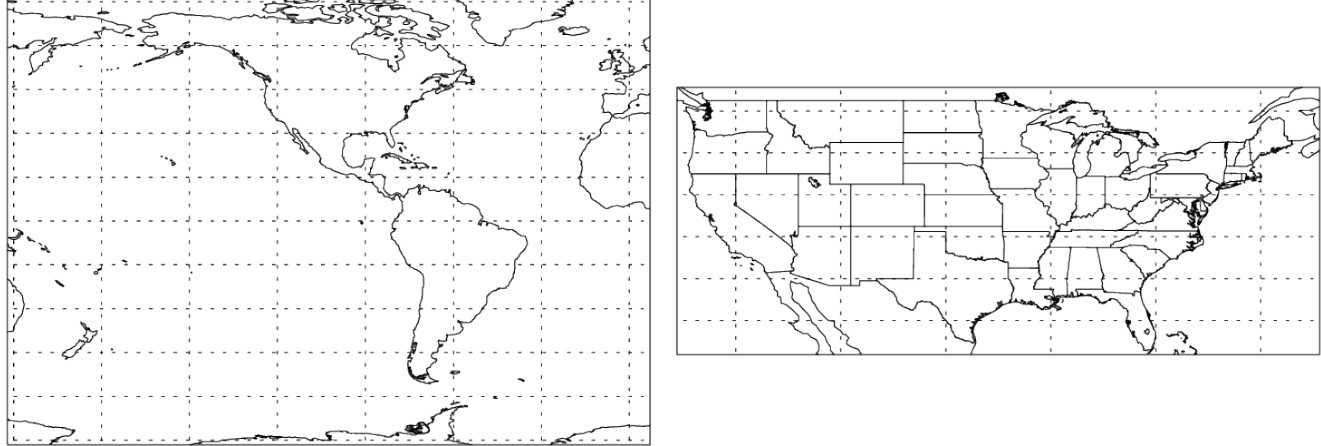

**Figure 3 - Spatial coverages of the 5375x3750 pixel domain of GridSat-GOES (left) and the 1500x650 pixel domain of GridSat-CONUS (right).**



**Table 1 – Summary of GOES data access for AREA files.**

|  | CLASS AREA files | Gridded-GOES |
|---|---|---|
| Format | AREA | netCDF-4 |
| Supported software languages | 1 (MCIDAS) | > 30 (supported at Unidata) |
| Navigation | Possible using 1000s lines of code | Stored directly in file |
| Calibration | IR: Information available at NESDIS site<br>Vis: Available in scientific journals | Stored directly in file |
| Temporal availability | Variable (1 to 30 minutes) | Regular (CONUS: 15 min, GOES: 1 hour) |
| Volume (1 month of data) | ~563 GB | ~45 GB |



**Table 2 – Comparison of GridSat-GOES and GridSat-CONUS with similar currently available datasets**

|  | Globally Merged IR | GridSat-B1 CDR | GridSat-GOES | GridSat-CONUS |
|---|---|---|---|---|
| Spatial Coverage | Global (60°N-60°S) | Global (70°N-70°S) | Western Hemisphere | Contiguous U.S. |
| Temporal coverage | 1998-present | 1980-present | 1994-present | 1994-present |
| Spatial resolution | ~4km | ~8 km | ~4km (0.04°) | ~4 km (0.04°) |
| Temporal resolution | 30 min | 3 hourly | Hourly | 15 minutes |
| Spectral coverage | 1 (IR) | 3 (IR, WV, VIS) | 8[1] (6 channels & 2 others) | 8[1] (6 channels & 2 others) |
| Climate quality intercalibration | None | IR | None | None |
| Volume (1 month) | 45 GB | 12 GB | 45 GB | 12 GB |

[1] See section 4.1 for a description of the channels in the variable list.



**Table 3 – List of duplicate files numbers by year. The percentages are relative to the number of scans in the year.**

| Year | Number of scans with duplicate files |
|------|--------------------------------------|
| 1994 | 448 (3%) |
| 1995 | 26,137 (41%) |
| 1996 | 15,830 (22%) |
| 1997 | 4743 (7.6%) |
| 1998 | 5 (0.01%) |
| 1999 | 37 (0.04%) |
| 2000 | 25 (0.02%) |
| 2001 | 41 (0.04%) |
| 2002 | 0 (0%) |
| 2003 | 153 (0.15%) |



**Table 4 – Number of files per satellite per year for GridSat-GOES.**

| Year | GOES-East | | | GOES-West | | | |
|------|--------|---------|---------|--------|---------|---------|---------|
|      | GOES 8 | GOES 12 | GOES 13 | GOES 9 | GOES 10 | GOES 11 | GOES-15 |
| 1994 | 2316 | | | | | | |
| 1995 | 8351 | | | 2324 | | | |
| 1996 | 8143 | | | 6794 | | | |
| 1997 | 7209 | | | 5903 | | | |
| 1998 | 8418 | | | 4526 | 3816 | | |
| 1999 | 8534 | | | | 8617 | | |
| 2000 | 8581 | | | | 8692 | | |
| 2001 | 8547 | | | | 8655 | | |
| 2002 | 8557 | | | | 8655 | | |
| 2003 | 2109 | 6511 | | | 8662 | | |
| 2004 | | 8610 | | | 8655 | | |
| 2005 | | 8585 | | | 8664 | | |
| 2006 | | 8581 | | | 4075 | 4586 | |
| 2007 | | 8277 | | | | 8647 | |
| 2008 | | 8204 | | | | 8676 | |
| 2009 | | 8287 | | | | 8353 | |
| 2010 | | 2449 | 6268 | | | 8664 | |
| 2011 | | | 8759 | | | 8040 | 609 |
| 2012 | | | 8179 | | | | 8695 |
| 2013 | | | 8280 | | | | 8757 |
| 2014 | | | 8755 | | | | 8231 |
| 2015 | | | 8759 | | | | 8753 |
| 2016 | | | 8779 | | | | 8782 |