# Peer review of "Gridded Satellite (GridSat) GOES and CONUS data"

_Earth System Science Data, 2018_

## Referee Comment (RC1) · Anonymous Referee #1 · 15 May 2018

Earth Syst. Sci. Data Discuss., https://doi.org/10.5194/essd-2018-33 Gridded Satellite (GridSat) GOES and CONUS data Kenneth R. Knapp and Scott L. Wilkins

General

This is a nice overview of a processed dataset to make access to historical GOES easier.

Technically, the legacy GOES is only the operational West satellite now, and will be replaced by the ABI by the time the paper is published. So, include some about the ABI, especially that it's on the Fixed Grid Format.

It turns out the line numbers started over on every page.

Specific

Line 8. GOES-A was launched in 1975, so say mid-1970s. Line 19. Not just weather, but the E is GOES is environmental, so volcanoes, fires, smoke, SST, LST, etc. Line 22. Prins should be Purdom. Line 24. Move the sentence about GOES-16 to the end of the paragraph, since GOES-17 will be at 137W, not 135W. Line 26. GOES Satellite is redundant. (Should be consistent, e.g., the following line 23)

Line 4. Add a reference for McIDAS. Line 28. What operational schedule supplies 1min imagery from legacy GOES? I think none. http://www.ospo.noaa.gov/Operations/GOES/west/srso.html So, add that the 1min images are special research scans and add a reference to two for the SRSOR run from GOES-14 during parts of 2012 to 2016.

Line 8. I trust you mean all GOES Imager channels, eg, not sounder. If so, this should be stated. Line 17. Change navigate to navigated. Line 17. Digital numbers seems redundant. How about digital counts, since they are integers. Line 18. Do you convert to reflectance, or reflectance factor (eg, no correction for the cosine of the zenith angle). Line 28. This is confusing. How about: There is the equivalent of an extra satellite . . .

Line 4. For completeness and clarity: There are generally about 120 GOES Imager scans per day; Line 5. How is optimal defined? Agreed that 60 and 15 min are reasonable cadences, but does that make them optimal? Line 11. Point out that an 18:00 UTC (not Z) image is the start time. Line 15. Are the delta times by line or by pixel along a line? Line 20. Missing 2 commas in NESDIS. Line 27. Need to define a reference temperature for a nedt. Line 27. There was an improvement in nedt between the early GOES (8 etc) and the 13-15 series. This should be noted. Line 30. So was the necessary boosting employed? I assume so, but this is un-clear. Is this a calibration, or an adhoc boost?

Line 4. What about the east-west over-sampling? Stated another way, were 3x3 original pixels used, or something else? Line 12. Can you do 15 min data from GOES-1

thru 7? Wasn't it more like every 30 min? Line 14. The ABI full disk scans are at least every 15 min, not 30 min. Line 15. Might note GOES-17 as well. Line 21. Increasing the spatial resolution may decrease the number of users, but I don't see how this is decreasing the usefulness. Again, with over-sampling the east-west ground sample distance is finer than 1 km. Line 25. Or somewhere, need to remind the readers about parallax. Line 26. What about stray light in the data?

Line 3. Consider readme files at the top level of the URLs. Line 8. Weren't there 2 periods, during GOES-13 outages, that GOES-14 was the operational satellite? This should be noted. Line 27. Where are the central wavelengths from? Channel 1 should have more precision. Channel 2 should be 3.9, not 3.75, channel 3 changed, during the series and hence it's just 6.7 um.

Line 14. Many users may not know what N-P are, just stick with the numbers to be consistent.

Figure 1. Note the projection.

Figure 2. This reviewer assumes a square root function has been applied to channel 1. If this is the case, it should be noted. It it's not the case, explain why the image seems so bright. Describe the color scheme for band 3. What s/w was used? Mcidas? If so, please state, at least in the acknowledgements. Explain in the caption the central wavelengths.

Figure 3. Again, what is the projection.

Table 1. Aren't there other systems that can read AREA files? Given their flat structure?

Table 2. It's stated that your new datasets start in 1994, but it should be noted this is September. Are the water vapor bands from all the GOES remapped to 4km? It might be confusing to say you have 6 channels, when you only have 5 at a time.

Table 3 is of limited interest for a user. Yet, this reviewer took one look at 18Z on the first day of the dataset from 1994 and how much stripping, missing lines. This was reflected

in artificially huge variance values. These type artifacts should be noted somewhere.

###
* * *
**ESSDD**

---

## Referee Comment (RC2) · Anonymous Referee #2 · 24 May 2018

Review of "Gridded Satellite (GridSat) GOES and CONUS data" Authors: Knapp and Wilkins Recommendation: Accept with minor revisions.

This paper describes a GOES-East and –West gridded satellite radiance product from NCEI which significantly increases availability and usefulness of this data (spanning 1994 – 2016). Historically, this data has been difficult to use in raw format and had a data volume which prohibits most scientific users. The authors have created a standardized product for both full disk and CONUS sectors, and included important metadata. Lead author Knapp has a proven record of rescuing geostationary data and making it easier to use for the community. This work represents another solid contribution. I found the paper to be well done and clearly presented with little need for revision, and recommend it for publication with just very minor revisions.

[Figure]

Specific comments: P2 L8: "from the archive" should be "from the NCEI archive"

P3 L14: "Next, simplified the scan times were selected". Fix grammar.

P3 L17: Once navigated.

P3 L24: It would be interesting for the authors to add some reason for why so many duplicate files, but not essential.

P4 L 4: Should be 120 images or sectors per day, not scans which also has the meaning of a scan line.

P4 L14: Was pleased to see the inclusion of delta_time for users which require this precision.

P5: Good job to include the list of future improvements.

Figure 1: List which channels are shown.

Table 2: Add a reference or be more specific for "Globally Merged IR", sounds too generic.

Table 3: Does this number of files mean all sectors? Please specify.

---

## Referee Comment (RC3) · Anonymous Referee #3 · 5 Jun 2018

page 3 line 17: Once navigated, ....

page 4 line 2: extremely valuable service!

page 4 line 16: Is this delta_time variable computed for each scan line? If not, how does one handle the computation of time for each pixel?

page 4 line 31: Can you say something about the calibration strategy. Was intersensor consistency or absolute radiometric accuracy your priority?

page 5 line 7: How do you handle the different resolutions. Do you average or sample the VIS to the 11 micron. I would suggest averaging and reporting the min and max. What about the 6.7 and 13.3 micron channels whose spatial resolution was larger than that of the 11 micron?

page 5 line 22: I repeat my comment that providing the mean, min and max VIS result within the nominal spatial resolution would be useful.

page 5 line 28: I would suggest including space for multiple calibrations.

---

## Author Comment (AC1) · 5 Jul 2018

**Response to referee comments on "Gridded Satellite (GridSat) GOES and CONUS data" by Kenneth R. Knapp and Scott L. Wilkins**

Kenneth R. Knapp and Scott L. Wilkins

**1. Introduction**

5    We welcome the comments from the referees provided in the forum. We will use the following format in the response below.

*Referee comments.*

     Author response.

       Change in manuscript

**2. Comments from Referee #1**

10    *General*

*This is a nice overview of a processed dataset to make access to historical GOES easier.*

*Technically, the legacy GOES is only the operational West satellite now, and will be replaced by the ABI by the time the paper is published. So, include some about the ABI, especially that it's on the Fixed Grid Format.*

     Actually, given the current issues with GOES-17, it is possible that GOES-15 continues to operate. We can't count

15      our eggs before they hatch.

*It turns out the line numbers started over on every page.*

*Specific*

*Line 8. GOES-A was launched in 1975, so say mid-1970s.*

       Added "mid" to the abstract.

20    *Line 19. Not just weather, but the E is GOES is environmental, so volcanoes, fires, smoke, SST, LST, etc.*

       Added "and other environmental conditions"

*Line 22. Prins should be Purdom.*

       Changed citation to Menzel and Purdom, 1994 and removed the Menzel and Prins citation.

*Line 24. Move the sentence about GOES-16 to the end of the paragraph, since GOES-17 will be at 137W, not 135W.*

25      This is meant as a general statement, so rewording the sentence should suffice.

       Changed "at 75" and "at 135" to "near 75" and "near 135" to imply that the values aren't exact.

       Removed "most" from line 22 since GOES-17 was also recently launched.

*Line 26. GOES Satellite is redundant. (Should be consistent, e.g., the following line 23)*

Agreed.

Removed satellite.

*Line 4. Add a reference for McIDAS.*

Included citation to Lazzara et al. (1999).

*Line 28. What operational schedule supplies 1-min imagery from legacy GOES? I think none. http://www.ospo.noaa.gov/Operations/GOES/west/srso.html So, add that the 1min images are special research scans and add a reference to two for the SRSOR run from GOES-14 during parts of 2012 to 2016.*

The statement that "the files can vary from 1 minute to 30 minutes, with a complex scan schedule that varies depending on the scan mode selected for the day" is accurate. Special rapid scan mode, when activated, creates havoc in attempting to make observations at regular intervals. It suffices that the parenthetical statement provides an example. We are not describing operational scan schedules but how the data appear to users accessing data in an archive. The data have varying temporal resolutions that can cause users to download data they don't need (which has happened based on NCEI archive comments).

Also, as stated in the manuscript, the GOES 14 rapid scan was not included in the table since it did not operationally replace GOES-East or GOES-West for a significant period.

No changes.

*Line 8. I trust you mean all GOES Imager channels, eg, not sounder. If so, this should be stated.*

Yes. GOES-Imager was meant. Reworded to avoid saying "GOES Imager images"

Changed to "The GOES Imager produces about 120 images per day"

*Line 17. Change navigate to navigated.*

Modified as suggested.

*Line 17. Digital numbers seems redundant. How about digital counts, since they are integers.*

Changed "numbers" to "counts"

*Line 18. Do you convert to reflectance, or reflectance factor (eg, no correction for the cosine of the zenith angle).*

Good catch. Visible data are not corrected for the cosine of the solar zenith angle, so the data are converted to reflectance, in keeping with terminology defined by Schaepman-Strub et al. (2006).

Added note to text that it is not corrected for solar angle so it not reflectance factor: "(note: visible data are not provided as reflectance factor, which would have a correction for the illumination geometry)"

*Line 28. This is confusing. How about: There is the equivalent of an extra satellite*

Change made as suggested.

*Line 4. For completeness and clarity: There are generally about 120 GOES Imager scans per day;*

Modified as suggested.

*Line 5. How is optimal defined? Agreed that 60 and 15 min are reasonable cadences, but does that make them optimal?*

Good point. We didn't mean to imply that a study was undertaken to determine the optimal number of observations per day. We merely meant that a nominal or regular interval was selected.

5          Reworded to "sectors store data at nominal 60 minute and 15 minute intervals"

*Line 11. Point out that an 18:00 UTC (not Z) image is the start time.*

The image represents a GridSat GOES image for 18:00, which is the nominal time. Since it is from a satellite at the GOES East position, then it starts at 17:45 UTC.

Changed caption of figure 1 to include "18:00 UTC image provided by a full disk scan that began at 17:45."

10   *Line 15. Are the delta times by line or by pixel along a line?*

The delta_time variable is calculated from the time of each scan line.

Changed to: "by storing the difference in time between the GridSat image time (17:45 and 18:00 in the previous example) and the time of the actual instrument scan as derived from the scan line information in the AREA files."

15   *Line 20. Missing 2 commas in NESDIS.*

Modified as suggested.

*Line 27. Need to define a reference temperature for a nedt.*

Changed discussion to "Brightness temperature noise is about 0.1 K at 300 K (Ellrod et al., 1998), which was improved to approximately 0.05 K at 300K for Imagers on GOES 13-15 Imagers (Zou et al., 2015)."

20   *Line 27. There was an improvement in nedt between the early GOES (8 etc) and the 13-15 series. This should be noted.*

Due to the need to reference something with GOES 13-15 numbers, I removed Wang et al and add a new citation. See change above.

*Line 30. So was the necessary boosting employed? I assume so, but this is un-clear. Is this a calibration, or an adhoc boost?*

Boosting? As stated in the manuscript, the correction is not ad hoc but follows Inamdar and Knapp.

25          I have attempted to clarify this by rewording it to "Therefore, visible channel data are calibrated in GridSat-GOES following Inamdar and Knapp (2015) to provide a temporally stable set of visible observations."

*Line 4. What about the east-west over-sampling? Stated another way, were 3x3 original pixels used, or something else?*

As stated in the manuscript: "This is calculated at the original satellite resolution." Since the original data provided

30   by the archive is oversampled, this is what is used in the calculation.

*Line 12. Can you do 15 min data from GOES-1 thru 7? Wasn't it more like every 30 min?*

Yes. The GOES data in the 1980s were mostly full disk scans that took 28 minutes. So the GridSat GOES wouldn't change while the CONUS would go to roughly every 30 minutes.

Changed "same temporal and spatial" to "similar"

*Line 14. The ABI full disk scans are at least every 15 min, not 30 min.*

Changed to 15 minutes

*Line 15. Might note GOES-17 as well.*

Changed to "GOES 16 and beyond" and named ABI in the text and cited Schmit et al.

5 *Line 21. Increasing the spatial resolution may decrease the number of users, but I don't see how this is decreasing the usefulness. Again, with over-sampling the east-west ground sample distance is finer than 1 km.*

There are multiple aspects of data usefulness. One significant aspect is ease of data access. Increasing data volume would make the data less useful in that users would need more storage to process the same amount of data. One important aspect of GridSat-CONUS/GOES was to decrease the burden on users downloading and storing the data.

10 To that end, 1 km data for the entire globe would be less useful. We agree that 1 km data has its applications.

*Line 25. Or somewhere, need to remind the readers about parallax.*

That is a good idea and we have included a new sentence in the caveat section.

Added "Furthermore, no adjustments were made for parallax error, which are cloud location errors at high view zenith angles (Vicente et al., 2002)."

15 *Line 26. What about stray light in the data?*

We can't provide caveats for all the GOES Imager issues. However, to this end we have added a sentence at the beginning of the section 4.4 on caveats.

Added "… some caveats should be considered prior to usage. First, users should become familiar with the limitations and quality of the source data: GOES Imager data."

20

*Line 3. Consider readme files at the top level of the URLs.*

We certainly can. As the access to this product grows, we will strive to meet user needs.

*Line 8. Weren't there 2 periods, during GOES-13 outages, that GOES-14 was the operational satellite? This should be noted.*

Yes. We note that occurs and that GOES 14 wasn't included in the Table because it wasn't a long time.

25 Added "it never assumes a role as either GOES-East or –West for a significant time"

*Line 27. Where are the central wavelengths from? Channel 1 should have more precision. Channel 2 should be 3.9, not 3.75, channel 3 changed, during the series and hence it's just 6.7 um.*

These are meant as general description of the channels on the many satellites.

Changed to "3.9"

30

*Line 14. Many users may not know what N-P are, just stick with the numbers to be consistent.*

Changed to "GOES 13-15"

*Figure 1. Note the projection.*

The data are stored in equal angle (0.04 deg) grids. So the projection is implicitly an equirectangular projection.

Added "Since data are stored on a 0.04° equal angle grid, simple displaying the data without transform, as in Figure 1, provides an equirectangular map projection." to section 2.3

*Figure 2. This reviewer assumes a square root function has been applied to channel 1. If this is the case, it should be noted. It it's not the case, explain why the image seems so bright.*

5            The data are sample imagery demonstrating what is possible with GridSat data. The visible data were scaled to show the large variability in the surface as well as the clouds.

Added to caption: "Visible data are enhanced to show detail across the dynamic range."

*Describe the color scheme for band 3.*

The data are sample imagery demonstrating what is possible with GridSat data.

10  *What s/w was used? Mcidas?If so, please state, at least in the acknowledgements.*

The software used for this process (reading, navigating and calibrating the AREA files) were IDL procedures written by the lead author.

No change necessary.

*Explain in the caption the central wavelengths.*

15            Central wavelengths are summarized in section 4.1.

Added to caption "Channel 6 data, central wavelengths are provided in section 4.1."

*Figure 3. Again, what is the projection.*

See above.

*Table 1. Aren't there other systems that can read AREA files? Given their flat structure?*

20            Yes. Some translators have been written. Do those constitute customer support of the software package?

*Table 2. It's stated that your new datasets start in 1994, but it should be noted this is September.*

Yes. But that would provide a resolution inconsistent with other rows. Each satellite has various start and end times as well as breaks where a different satellite was used to cover an outage. These all can't be listed. The point of the Table is to show completeness or lack thereof.

25  *Are the water vapor bands from all the GOES remapped to 4km?*

Yes.

*It might be confusing to say you have 6 channels, when you only have 5 at a time.*

It might be confusing at first, but that is why we mention it in the caption of Figure 3 as well as the description of the variables in section 4.1.

30  *Table 3 is of limited interest for a user.*

Not for someone who tries to download a large amount of data from the archive. In this case, this table shows the extra work that will be needed to understand and work with the data. Thus, this table shows some of the work that has been done to facilitate use of GOES data using GridSat GOES.

*Yet, this reviewer took one look at 18Z on the first day of the dataset from 1994 and how much stripping [sic], missing lines. This was reflected in artificially huge variance values. These type artifacts should be noted somewhere.*

**3. Comments from Referee #2**

5    *This paper describes a GOES-East and –West gridded satellite radiance product from NCEI which significantly increases availability and usefulness of this data (spanning 1994 – 2016). Historically, this data has been difficult to use in raw format and had a data volume which prohibits most scientific users. The authors have created a standardized product for both full disk and CONUS sectors, and included important metadata. Lead author Knapp has a proven record of rescuing geostationary data and making it easier to use for the community. This work represents another solid contribution.*

10   *I found the paper to be well done and clearly presented with little need for revision, and recommend it for publication with just very minor revisions.*

*Specific comments:*

*P2 L8: "from the archive" should be "from the NCEI archive"*

Modified as suggested.

15   *P3 L14: "Next, simplified the scan times were selected". Fix grammar.*

Changed to "Next, specific scan times were selected"

*P3 L17: Once navigated.*

Changed as suggested.

*P3 L24: It would be interesting for the authors to add some reason for why so many duplicate files, but not essential.*

20   This is kind of described at the start of section 2.1. To make it a bit more clear, we have summarized the opening paragraph.

Added "This produced duplicate files of the same image."

*P4 L 4: Should be 120 images or sectors per day, not scans which also has the meaning of a scan line.*

Good point. Rewording was required to avoid saying "GOES Imager images"

25   Changed to "The GOES Imager produces about 120 images per day"

*P4 L14: Was pleased to see the inclusion of delta_time for users which require this precision.*

*P5: Good job to include the list of future improvements.*

*Figure 1: List which channels are shown.*

Modified caption to read in part "Sample of infrared window GridSat"

30   *Table 2: Add a reference or be more specific for "Globally Merged IR", sounds too generic.*

Added as suggested

*Table 3: Does this number of files mean all sectors? Please specify.*

The numbers here provide the number of GridSat files per year. Since GridSat-GOES is hourly, then the optimal number is 8760 files (8784 for leap years). I've added some explanatory text to help with the interpretation of this table

5     Added "The most possible in a year is 8760 files at 24 files per day (and 8784 in a leap year). The variation in numbers show gaps in the record: large deviations from 8760 in the 1990s with more stable and relatively few missing files in the more recent years."

**4. Comments from Referee #3**

*page 3 line 17: Once navigated, ....*

    Changed as suggested.

10   *page 4 line 2: extremely valuable service!*

*page 4 line 16: Is this delta_time variable computed for each scan line? If not, how does one handle the computation of time for each pixel?*

    The original text wasn't clear.

    Modified to read "by storing the difference in time between the GridSat image time (17:45 and 18:00 in the

15     previous example) and the time of the actual instrument scan as derived from the scan line information in the AREA files."

*page 4 line 31: Can you say something about the calibration strategy. Was intersensor consistency or absolute radiometric accuracy your priority?*

    The correction is primarily an attempt to produce temporally stable observations (to correct for sensor degradation).

20

*page 5 line 7: How do you handle the different resolutions. Do you average or sample the VIS to the 11 micron. I would suggest averaging and reporting the min and max.*

    The data were sampled and data were not averaged as stated in section 2.3. So the footprints are at different resolutions. This is why a visible channel variability was included to provide some measure of representativeness of

25     the 1 km resolution at a larger scale. An average could certainly be applied and provided with other statistics (e.g., min and max), but would significantly increase data volume. Such a change would be warranted given enough user feedback.

    No change necessary since it is explained in section 2.3.

*What about the 6.7 and 13.3 micron channels whose spatial resolution was larger than that of the 11 micron?*

30     Again, data were sampled.

*page 5 line 22: I repeat my comment that providing the mean, min and max VIS result within the nominal spatial resolution would be useful.*

We appreciate the suggestion and will consider doing so based on input from users as we receive information.

*page 5 line 28: I would suggest including space for multiple calibrations.*

We certainly will work to include this in a future revision should users request it.

**References**

5  Lazzara, M. A., Benson, J. M., Fox, R. J., Laitsch, D. J., Rueden, J. P., Santek, D. A., Wade, D. M., Whittaker, T. M., and Young, J. T.: The Man computer Interactive Data Access System: 25 Years of Interactive Processing, Bulletin of the American Meteorological Society, 80, 271-284, 1999.

Schaepman-Strub, G., Schaepman, M. E., Painter, T. H., Dangel, S., and Martonchik, J. V.: Reflectance quantities in optical remote sensing—definitions and case studies, Remote Sensing of Environment, 103, 27-42, 2006.

10